# Highly sensitive CE-ESI-MS analysis of *N*-glycans from complex biological samples

Guinevere S.M. Lageveen-Kammeijer [1,3], Noortje de Haan [1,3], Pablo Mohaupt[1], Sander Wagt[1], Mike Filius [1,2], Jan Nouta[1], David Falck [1] & Manfred Wuhrer [1]

The in-depth, high-sensitivity characterization of the glycome from complex biological samples, such as biofluids and tissues, is of utmost importance in basic biological research and biomarker discovery. Major challenges often arise from the vast structural diversity of glycans in combination with limited sample amounts. Here, we present a method for the highly sensitive characterization of released *N*-glycans by combining a capillary electrophoresis-electrospray ionization-mass spectrometry (CE-ESI-MS) approach with linkage-specific derivatization of sialic acids and uniform cationic reducing end labelling of all glycans. This method allows the analysis of glycans at the attomole level, provides information on sialic acid isomers and enables the in-depth characterization of complex samples, even when available in minute amounts.

[1] Leiden University Medical Center, Center for Proteomics and Metabolomics, Postbus 9600, 2300 RC Leiden, The Netherlands. [2] TU Delft, BioNanoScience Department, van der Maasweg 9, 2629 HZ Delft, The Netherlands. [3] These authors contributed equally: Guinevere S.M. Lageveen-Kammeijer, Noortje de Haan. Correspondence and requests for materials should be addressed to G.S.M.L.-K. (email: g.s.m.kammeijer@lumc.nl)

Glycosylation is known as one of the most complex post-translational protein modifications and is involved in many biological processes, including cellular differentiation and antibody-receptor interactions[1]. Characterizing the glycome of a complex biological sample can provide essential information on proteoform heterogeneity and changes in the glycan biosynthesis pathway. A major challenge posed for the glycomic analysis of biological samples is the immense complexity of the glycome. This complexity is reflected in the vast number of different structures present, including glycan isomers, and in the broad dynamic range of glycan abundancies[2]. Especially in the case of precious clinical samples, methods for the characterization of minor glycoforms and glycans originating from low-concentration glycoproteins are scarce.

Techniques currently used for the sensitive and in-depth characterization of released N-glycans involve, amongst others, capillary gel electrophoresis (CGE) coupled to laser-induced fluorescence (LIF) detection after aminopyrene trisulfonate (ATPS) labelling of the reducing ends of the glycans[3] and nano liquid chromatography (LC)-mass spectrometry (MS) using porous graphitized carbon (PGC) as stationary phase[4,5]. While the former approach shows the ability to perform a sensitive and in-depth characterization of glycan isomers in a high-throughput manner, latter approach proves particularly valuable in the case complex samples with unknown glycans are analysed, allowing their structural elucidation using negative-mode tandem mass spectrometry.

Here, we achieve high-sensitivity profiling of N-glycans by combining a capillary electrophoresis (CE) separation method with positive ion mode electrospray ionization (ESI)-MS[6]. The sheathless interface between the CE and the ESI-MS provides low-flow ($<10$ nL min$^{-1}$) nano-ESI conditions, resulting in maximum intensities due to efficient droplet desolvation and ionization of the analytes[7,8]. To enable effective electrophoretic migration of all N-glycans, sialylated species are neutralized and all species are uniformly charged by the reducing end label Girard's reagent P (GirP). Next to neutralization, the sialic acid derivatization strategy allows to differentiate sialic acid isomers without the use of MS fragmentation, which can provide important information on biological processes, such as cancer development[9].

## Results

### Linkage-specific sialic acid neutralization

To neutralize all glycans and enable the linkage-specific differentiation of sialic acids, we developed a method which results in the ethyl esterification of α2,6-linked sialic acids (derivatized sialic acid mass: 319.127 Da; Fig. 1a) and amidation with ammonia of α2,3-linked sialic acids (derivatized sialic acid mass: 290.111 Da; Fig. 1b) in a facile two-step, one-pot reaction. While, various methods are known for the sialic acid linkage-specific derivatization of glycans[10–14], the majority of these methods are based on the lactonization of α2,3-linked sialic acids[13]. Lactones are, however, prone to hydrolysis during sample storage or further sample processing[11,15,16]. These stability issues were recently overcome for on-tissue derivatization, by a two-step approach in which, during the second step, the lactones are hydrolysed and stably amidated[11]. In the current method, a similar approach was implemented for released glycans in solution, resulting in stable derivatives. As compared with previous approaches[17,18], this method is simpler as it relies on a one-pot reaction and comes with a single hydrophilic interaction liquid chromatography (HILIC)-solid phase extraction (SPE) clean-up step to remove excess reagents. The method shows excellent linkage specificity and results in $100.0\% \pm 0.02\%$ (±standard deviation) of amide formation and $99.1\% \pm 0.3\%$ of ethyl ester formation for 3′-sialyllactose ($n = 4$) and 6′-sialyllactose ($n = 4$), respectively ($m/z$ 775.286 and 804.301; detected as sodium adducts by matrix assisted laser desorption/ionisation (MALDI)-time of flight (TOF)-MS; Supplementary Fig. 1). In addition, even when performed on complex samples, such as N-glycans released from total human plasma (TPNG), minimal by-products were observed (Supplementary Fig. 2).

### Permanent cationic labelling of the N-glycan reducing ends

To enable electrophoretic migration of the neutralized N-glycans[15,16], we introduced the hydrazide GirP, carrying a permanent positive charge, to all species via reducing end labelling (Fig. 1c). The one-step, clean-up free protocol allowed the direct injection of the labelled N-glycans into the CE-ESI-MS. The applicability of hydrazide labels has been shown before in the analysis of released N-glycans by LC-ESI-MS and MALDI-TOF-MS methods, enhancing ionization and separation efficiency[19]. As compared with previous reports[19,20], we optimized the efficiency and manageability of the hydrazide labelling by using ethanol as solvent, limiting the amount of water in the reaction, increasing the GirP concentration to 50 mM and reducing the incubation time to 1 h. This resulted in a labelling efficiency of $87\% \pm 4\%$ (as determined by MALDI-TOF-MS; $n = 3$; Supplementary Table 1, Supplementary Fig. 3). Notably, because the sialic acid derivatization neutralized all N-glycans, the subsequent GirP labelling resulted in a uniform charge on all species enabling efficient ionization in positive-ionization mode.

### Glycoform characterization, sensitivity and repeatability

To further improve the sensitivity of the CE-ESI-MS platform, we implemented a dopant enriched nitrogen (DEN)-gas at the interface between the CE and the ESI-MS to boost ionization of GirP-labelled glycans. Recently, we showed a substantial gain in sensitivity for the analysis of glycopeptides using DEN-gas[8]. In the current study, the detection of the twenty most abundant GirP-labelled N-glycans in plasma with DEN-gas was compared with their detection without DEN-gas. Overall, we observed less background and an, on average, 3.3-fold enhancement of the signal-to-noise ratios (S/N), while the relative profiles stayed the same (Supplementary Figs. 4 and 5), which is in correspondence with the previous study[8].

The absolute sensitivity of the overall method was assessed by analysing the glycan standards H3N4 and H5N4. A starting amount of 5 fmol of both glycans, in a sample volume of 3 μL, was sufficient for their detection and reliable relative quantification (H3N4: S/N = 49 ± 18; H5N4: S/N = 146 ± 52; Fig. 2a–c, Supplementary Table 2; $n = 3$). The starting amount of 5 fmol corresponded to an absolute amount of 20 amol (43 nL corresponding to 1/250 of the sample) injected into the CE-ESI-MS system.

The performance of the fully optimized method for complex biological samples was demonstrated on TPNG, which is a sample with numerous glycoforms present in abundances that range over several orders of magnitude (Fig. 2d–f)[12,21]. In total, 167 N-glycan compositions were detected after the injection of the equivalent of 0.1 nL PNGase F treated plasma—derived from a starting amount of 0.2 μL PNGase F treated plasma—into the CE-ESI-MS setup. N-glycans were detected, and compositions were assigned, based on accurate mass and isotopic pattern. In addition, 82 glycoforms were confirmed by tandem mass spectrometry via collision-induced dissociation (CID; Fig. 3; Supplementary Fig. 6, Supplementary Data 1). Distinct diagnostic ions were observed in the MS/MS data of the GirP-labelled N-glycans (Supplementary Table 3), e.g. signals at $m/z$ 656.251$^{1+}$ and $m/z$ 685.260$^{1+}$ are B-ions indicating a full antenna loss with an amidated (α2,3-linked) or an ethyl esterified (α2,6-linked) sialic acid, respectively. In the case of a fucosylated, α2,3-linked

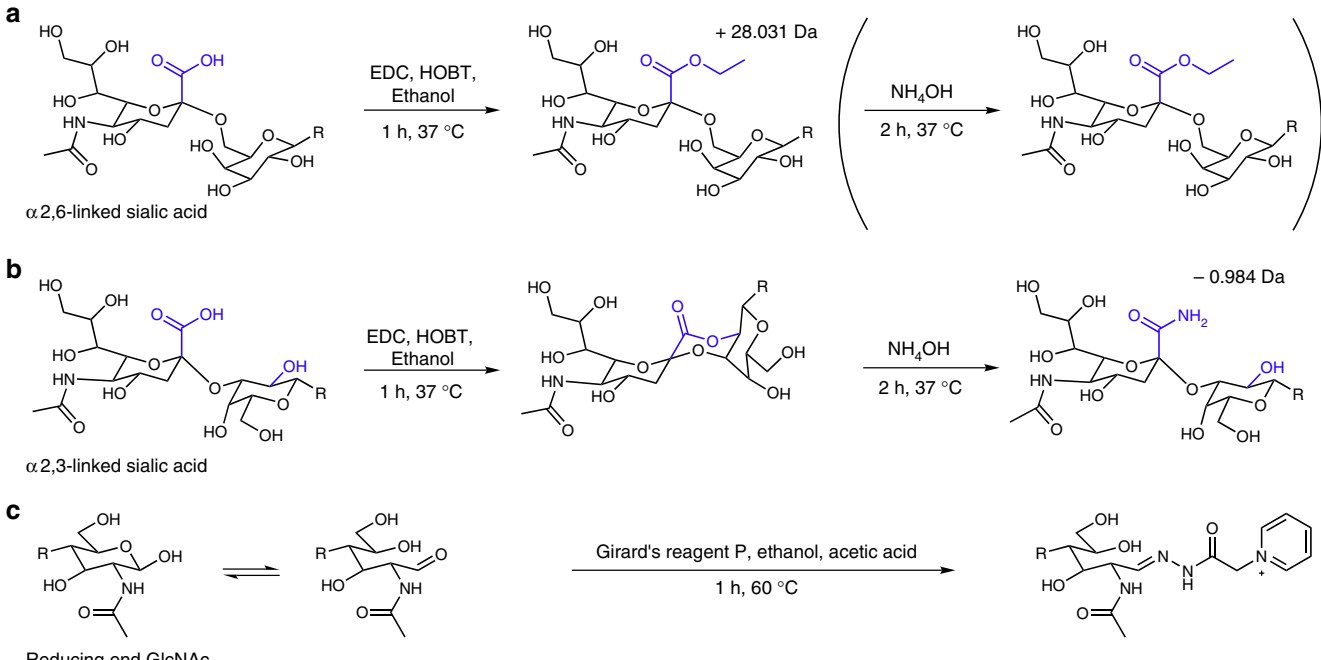

**Fig. 1** Reaction scheme for the linkage-specific derivatization of sialic acids and subsequent GirP labelling. **a** The α2,6-linked sialic acid forms an ethyl ester during the first step of the reaction and remains stable throughout. **b** The α2,3-linked sialic acid initially loses water forming a lactone. After the addition of ammonia, the lactone ring is opened and a stable amide is formed. **c** The reducing ends of all *N*-glycans are labelled with Girard's reagent P

sialylated antenna, a B-ion at $m/z$ 802.309$^{1+}$ was observed, indicating that fucosylation of sialylated antennae in TPNG occurs in the context of sialyl Lewis-type structures such as sialyl Lewis X[22,23]. Furthermore, the Y-ion detected at $m/z$ 355.162$^{1+}$ corresponds to the GirP label with an *N*-acetylglucosamine, while the ion at $m/z$ 501.220$^{1+}$ indicates the presence of core fucosylation [GirP + *N*-acetylglucosamine + fucose]$^{+}$. Bisection was identified by a Y-ion present at $m/z$ 923.373$^{1+}$, or in combination with a core fucose at $m/z$ 1069.431$^{1+}$.

As compared with previous studies assessing the TPNG in a sialic acid linkage-specific manner, we here report a higher number of unique glycan compositions (Supplementary Data 1 and 2)[5,12,16,24–28]. The lowest abundant glycoform that could be quantified (S/N > 9) had a relative abundance of 0.007% ± 0.003% (H8N7F1S$_{2,3}$4 at low attomole detection level; $n$ = 9; Supplementary Table 4 and Supplementary Fig. 7), highlighting the broad dynamic range of the platform. Only minimal (<1%) cation adduct formation was observed, resulting in simple spectra, with high sensitivity in both MS and tandem MS (Supplementary Fig. 8). While similar sensitivities and dynamic ranges were reached before with a targeted method[5], here we used an untargeted method, better suitable for discovery research. Furthermore, we found a very good overall method repeatability and intermediate precision, based on the relative quantification of the 118 glycoforms with a S/N > 9 (Fig. 4 and Supplementary Data 1). The preparation, analysis and relative quantification of three TPNG samples per day on three consecutive days resulted in a median RSD of the 20 most abundant *N*-glycans (accounting for about 80% of the *N*-glycans quantified) of 6.8% for the repeatability (intraday variability day 1), while the interday variability of day 1–3 showed a median RSD of 9.4%.

**Conclusion.** We present a highly sensitive CE-ESI-MS(/MS) platform for the in-depth analysis of *N*-glycans at the attomole level. Applying the method on a complex, biological sample, like

the *N*-glycans released from total human plasma, 167 *N*-glycan compositions were identified, including different sialic acid linkage variants. We believe that the developed platform is of interest for the in-depth analysis of low abundant glycans in complex biological matrices. Furthermore, the glycosylation of proteins too low abundant, or samples too precious, to be measured with conventional methods becomes accessible with this method.

## Methods

**Materials and reagents**. Ethanol (EtOH; cat. nr. 1.00983.1000), trifluoroacetic acid (TFA; cat. nr. 1.08178.0050), disodium hydrogen phosphate dihydrate (Na$_2$HPO$_4$·2H$_2$O; cat. nr. 1197530250), potassium dihydrogen phosphate (KH$_2$PO$_4$; cat. nr. 1048730250), sodium chloride (NaCl; cat. nr. 106404) and glacial acetic acid (HAc; cat. nr. 1000631000) were purchased from Merck (Darmstadt, Germany). Recombinant peptide-*N*-glycosidase F (PNGase F; cat. nr. 11365177001) was acquired from Roche Diagnostics (Mannheim, Germany). Ammonium acetate (AmAc; cat. nr. A2706), 2-aminobenzamide (2-AB; cat. nr. A89804), 2-picoline borane (2-PB; cat. nr. 654213), 1-hydroxybenzotriazole (HOBt; cat. nr. 54802) hydrate, dimethyl sulfoxide (DMSO; cat. nr. D8418), 40% dimethylamine in water (cat. nr. 426458), super-DHB (cat. nr. 50862), Nonidet P-40 substitute (NP-40; cat. nr. M158), 50% sodium hydroxide (NaOH; cat. nr. 71686), sodium dodecyl sulfate (SDS; cat. nr. L3771), and 28% ammonium hydroxide (NH$_4$OH; cat. nr. 221228) were purchased from Sigma Aldrich (Steinheim, Germany). 1-Ethyl-3-(3-(dimethylamino)propyl) carbodiimide (EDC; cat. nr. 24810) hydrochloride was obtained from Fluorochem (Hadfield, UK). The ultrapure deionized water (MQ) used in this study was generated from a Q-Gard 2 system (Millipore, Amsterdam, Netherlands), maintained at ≥18 MΩ. LC-MS Ultra water (H$_2$O; cat. nr. 00232141B1BS) was purchased from Honeywell (Morris Plains, NJ) and HPLC SupraGradient acetonitrile (MeCN; cat. nr. 1203502) was purchased from Biosolve (Valkenswaard, Netherlands). 1-(Hydrazinocarbonylmethyl)pyridinium chloride (Girard's Reagent P; GirP; cat. nr. G0030) was purchased from TCI Development Co. Ltd. (Tokyo, Japan). A standard plasma pool from healthy donors was obtained from Affinity Biologicals (Visucon-F; Ancaster, ON, Canada; cat. nr. FRNCP0125; lot nr. 000752FCP). Two oligosaccharide standards, 3′-sialyllactose sodium salt and 6′-sialyllactose sodium salt, were obtained from Carbosynth (Compton, UK; cat. nr.; OS04397 and OS04398; lot nr. OS043971301 and OS043981301, respectively) and two glycan standards (H3N4 and H4N5; cat. nr. CN-NGA2-20u and CN-NA2-20u; lot nr. B75I04 and B65Q02, respectively) were kindly provided by Ludger Ltd. (Abingdon, UK). A 10× phosphate-buffered saline solution (10× PBS, pH 7.2) was prepared in-house, containing 57 g L$^{-1}$ Na$_2$H-PO$_4$·2H$_2$O, 5 g L$^{-1}$ KH$_2$PO$_4$ and 85 g L$^{-1}$ NaCl.

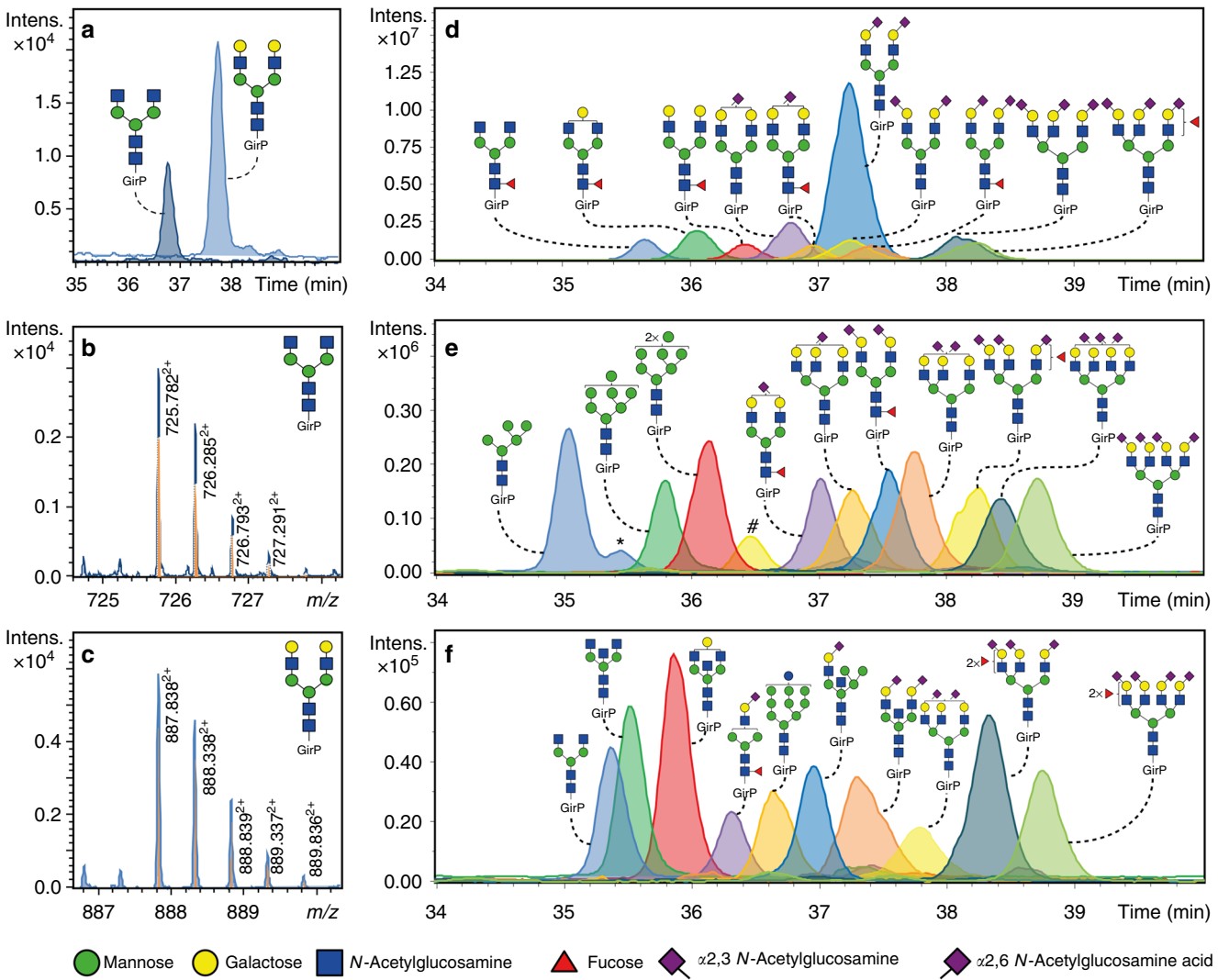

**Fig. 2** Extracted ion electropherograms of derivatized and labelled *N*-glycans. **a** Extracted ion electropherograms of two *N*-glycan standards (H3N4 and H5N4) after injection of a final amount of 20 amol. **b, c** Illustration of the obtained isotopic pattern and mass accuracy (blue trace) for the doubly charged species H3N4 and H5N4 compared with the theoretical isotopic pattern (orange trace). **d** Extracted ion electropherograms of the 10 most abundant *N*-glycans (>2% relative abundance) present in TPNG, **e** 10 medium abundant *N*-glycans (between 0.5–1.0% relative abundance) and **f** 10 low abundant *N*-glycans (<0.25% relative abundance). Separation was achieved using a bare-fused silica capillary after a dynamic neutral coating. *Indicates in-source decay of the H6N2. # Shows the presence of doubly charged species of H5N5F1 of which the *m/z* values overlap with the ones of the triply charged species of H6N5F1S$_{2,3}$2S$_{2,6}$1. Blue square: *N*-acetylglucosamine, green circle: mannose, yellow circle: galactose, red triangle: fucose, right pointing pink diamond: α2,6-linked *N*-acetylneuraminic acid, left pointing pink diamond: α2,3-linked *N*-acetylneuraminic acid, GirP: Girard's reagent P label

**N-Glycan release.** *N*-glycans were released from all plasma proteins (TPNG) by mixing 100 μL of plasma with 200 μL of 2% SDS followed by a 10 min incubation at 60 °C. Subsequently, a 200 μL mixture of 2.5× PBS, 2% NP-40 and 10 U PNGase F was added and the sample was incubated at 37 °C for 17 h.

**Sialic acid neutralization and purification of N-glycans.** Sialic acid linkage-specific derivatization by ethyl esterification was performed as described previously[12]. Briefly, 1 μL of released plasma *N*-glycans (containing the released glycans from 0.2 μL of plasma) was added to 20 μL ethyl esterification reagent (250 mM EDC and 250 mM HOBt in EtOH) and incubated for 1 h at 37 °C. Subsequently, 20 μL MeCN was added and *N*-glycans were purified by cotton HILIC SPE as described before[12,29]. Samples were eluted in 10 μL MQ. In addition, an amidation step was introduced in the protocol, for the robust stabilization of the α2,3-linked sialic acids, by adding 4, 6 or 8 μL of 28% NH$_4$OH to the reaction mixture after 1 h incubation. The addition of the ammonia was followed by an incubation step of 2 h at 37 °C. After incubation, 24, 26 or 28 μL MeCN was added to the mixture and the *N*-glycans were purified by cotton HILIC SPE, with an elution in 10 μL MQ. The optimized ethyl esterification protocol with amidation step (EEA) used 4 μL of 28% NH$_4$OH solution. Furthermore, another option for the linkage-specific derivatization of the sialic acids was investigated, in which the sialic acids were differentially modified by double amidation (DA), as described

previously[11]. Briefly, 1 μL of released plasma *N*-glycans was added to 20 μL dimethylamidation reagent (250 mM dimethylamine, 250 mM EDC and 500 mM HOBt and in DMSO) and incubated for 1 h at 60 °C. An additional incubation followed of 2 h at 60 °C after the addition of 8 μL 28% NH$_4$OH. Eighty microlitres of MeCN was added to the samples and the derivatized *N*-glycans were purified by cotton HILIC SPE, with elution in 10 μL MQ.

**Sialic acid neutralization and purification of sialyllactose.** The linkage specificity of the amidation step following the ethyl esterification protocol was validated on 3′-sialyllactose and 6′-sialyllactose standards. Prior to sialic acid derivatization, the standards were labelled with 2-AB. In total, 30 μL 50 mg mL$^{-1}$ sialyllactose was incubated with 30 μL 24 mg mL$^{-1}$ 2-AB and 53.5 mg mL$^{-1}$ 2-PB in 7.5%/92.5% HAc/DMSO for 2 h at 60 °C. MeCN was added to the samples in a final concentration of 90% and the samples were purified by cotton HILIC SPE as described previously[12,29], with the exception that 90% MeCN washing solutions were used instead of 85% MeCN washing solutions. Samples were eluted in 10 μL water, of which 1 μL was subjected to either ethyl esterification or EEA as described above. After incubation, 20, 24, 26 or 28 μL MeCN was added to the mixtures and the sialyllactoses were purified by cotton HILIC SPE, using 90% MeCN washing solutions and an elution in 10 μL MQ.

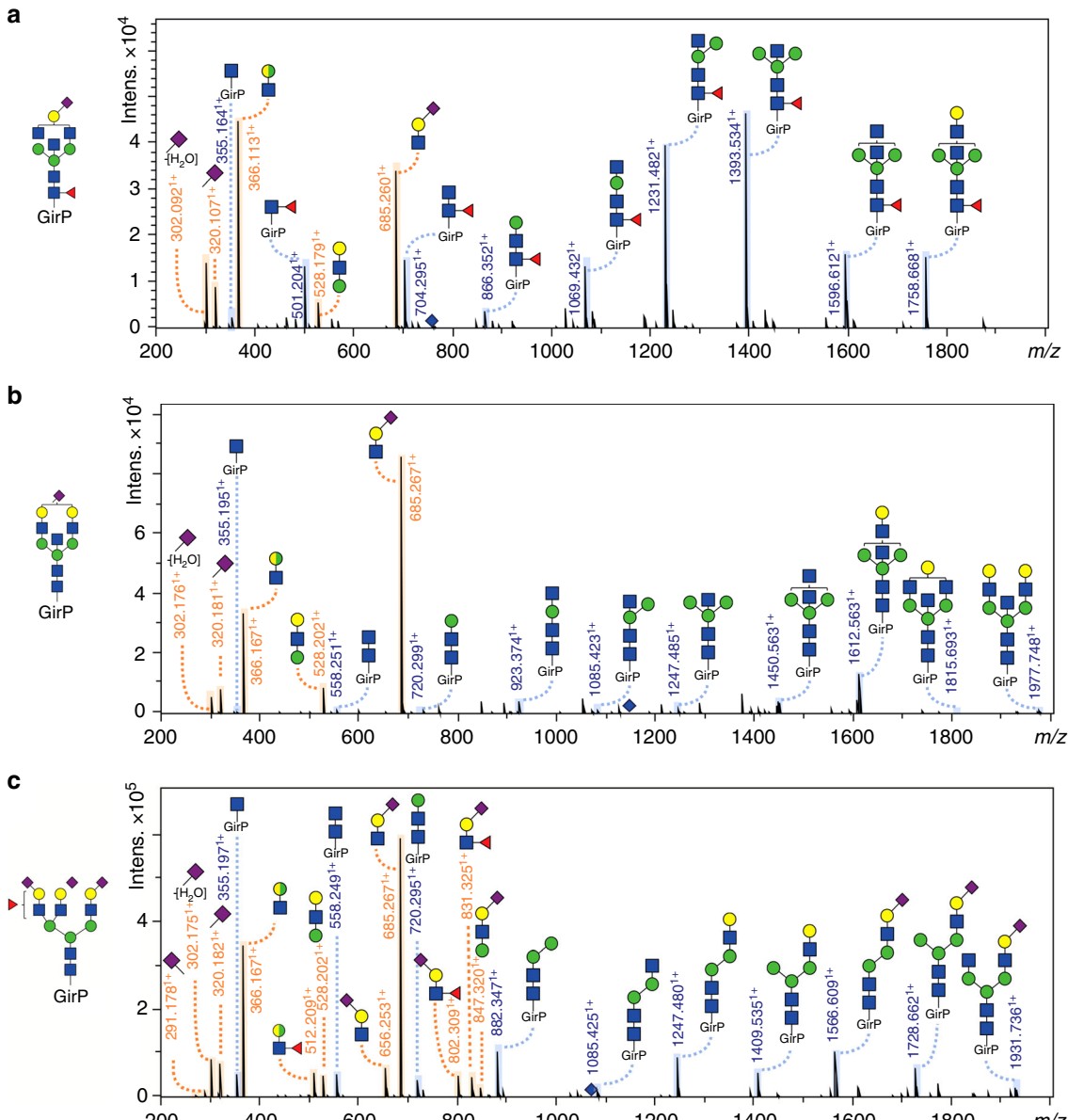

**Fig. 3** CE-ESI-MS/MS fragmentation spectra of derivatized and labelled *N*-glycans from TPNG. The following precursor ions were fragmented:
**a** 760.960$^{3+}$ (H4N5F1S$_{2,6}$1), **b** 1148.939$^{2+}$ (H5N5S$_{2,6}$1) and **c** 1072.076$^{3+}$ (H6N5F1S$_{2,3}$1S$_{2,6}$2). The Y-ion fragmentation pattern is annotated in blue and the B-ion fragmentation pattern is annotated in orange. Blue diamond: precursor ion, blue square: *N*-acetylglucosamine, green circle: mannose, yellow circle: galactose, red triangle: fucose, right pointing pink diamond: α2,6-linked *N*-acetylneuraminic acid, left pointing pink diamond: α2,3-linked *N*-acetylneuraminic acid, GirP: Girard's reagent P label. *The presence of an antennary fucose in combination with an α2,6-linked sialic acid is likely, for a large part, due to fucose migration during MS analysis[36]

**Permanent cationic labelling of the *N*-glycan reducing end.** After sialic acid derivatization, either by EEA or DA, released *N*-glycans from plasma were labelled at the reducing end by GirP. Initial reaction conditions were as follows, 5 μL of sialic acid-derivatized, HILIC-purified *N*-glycans were mixed with 95 μL GirP reagent (15 mM GirP in 85% EtOH, 10% HAc and 5% MQ) and incubated for 2 h at 60 °C. After incubation, 105 μL MeCN was added and cotton HILIC SPE was performed as described above, using 85% MeCN washing solutions and 10 μL elution volume. Optimization of the protocol (Supplementary Table 1) included decreasing the percentage of MQ in the reagent (0, 5, 15 or 25%), while keeping the other factors constant. Using 0% MQ in the reagent, the concentration of GirP was increased (15, 22, 33 or 50 mM). With 0% MQ and 50 mM GirP in the reagent, the sample volume was evaluated (5 μL or dried) and on the dried sample, the reagent volume was decreased (95, 50, 20, 10, 5 or 2 μL). Finally, the incubation time was evaluated (1 or 2 h). The optimized reaction conditions included a drying step of the *N*-glycan sample prior to adding 2 μL of GirP reagent (50 mM GirP in 90% EtOH and 10% HAc) and 1 h incubation at 60 °C. After incubation, the samples were dried by vacuum concentration at 60 °C and dissolved in 10 μL MQ for CE-ESI-MS analysis or 33.3 μL 85% MeCN for HILIC purification and MALDI-TOF-MS analysis.

**CE-ESI-MS(/MS) Analysis.** All experiments were performed on a 91 cm long bare-fused silica capillary (30 μm internal diameter and 150 μm outer diameter) using a CESI 8000 system (Sciex, Framingham, MA). Prior to usage, the capillary was conditioned by immersing the spray tip in MeOH while the separation and conductive lines were rinsed with MeOH at 100 psi for 10 min and 3 min, respectively. Subsequently, the tip was immersed in H$_2$O and the separation line was rinsed at 100 psi for 10 min consecutively with H$_2$O, 0.1 M NaOH, 0.1 M HCl, H$_2$O and the background electrolyte (BGE; 10% HAc), followed by a final rinsing step of the conductive line for 3 min with the BGE at 100 psi. After the conditioning steps, the capillary was coated with Ultratrol dynamic pre-coat LN (UT; Target Discovery, Palo Alto, CA) as described by Kohler et al.[30] with some small adjustments. Briefly, using 29 psi throughout, the separation line was coated by rinsing consecutively for 10 min with MeOH, H$_2$O, 0.1 M HCl and H$_2$O, 15 min rinsing with 1 M NaOH followed by 15 min rinsing with H$_2$O, finally a 30 min rinsing was performed with UT. After the coating procedure, the capillary was rinsed for 30 min with the BGE at 29 psi. Prior to each analysis the capillary was rinsed with 1 M NaOH (3 min), H$_2$O (4 min), UT (4 min) and BGE (3 min) all at 29 psi. To ensure the removal of the UT from the outlet of the capillary, an

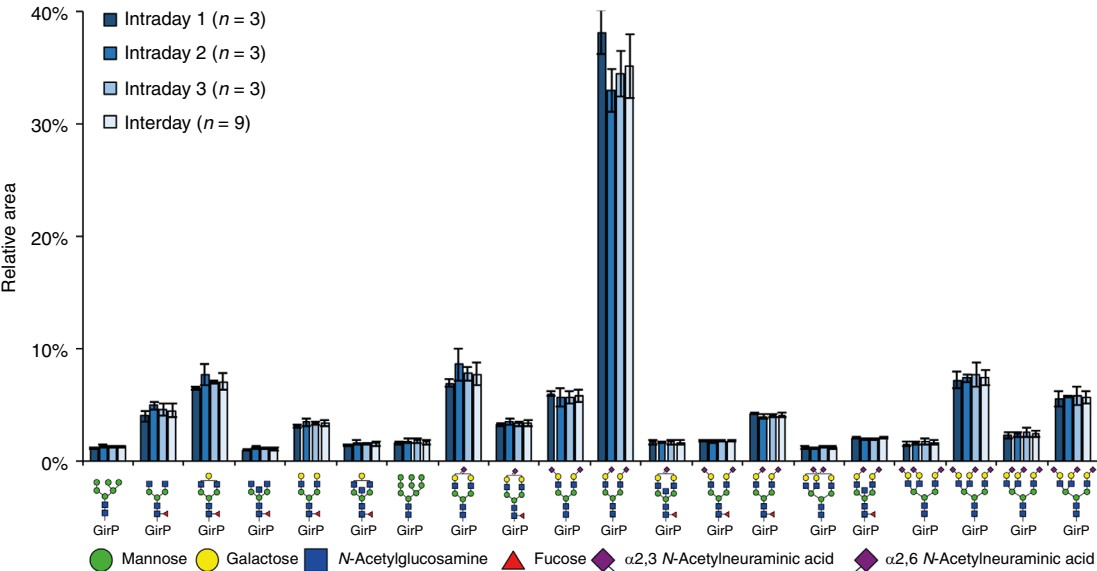

**Fig. 4** Inter- and intraday repeatability of the TPNG analysis by CE-ESI-MS. The released *N*-glycans were analysed after sialic acid derivatization and reducing end labelling with GirP. The relative abundances of the 20 most abundant glycoforms are displayed (for a complete overview, see Supplementary Fig. 7), showing a median RSD of 6.8% within technical replicates form the same day and a median RSD of 9.4% over 3 days. To enable automated peak integration, the electropherograms were aligned prior to data extraction[32]. The relative glycan profiles and their variation were similar before and after alignment (Supplementary Figs. 10 and 11). Blue square: *N*-acetylglucosamine, green circle: mannose, yellow circle: galactose, red triangle: fucose, right pointing pink diamond: α2,6-linked *N*-acetylneuraminic acid, left pointing pink diamond: α2,3-linked *N*-acetylneuraminic acid, GirP: Girard's reagent P label. Error bars represent the standard deviation (intraday *n* = 3, interday *n* = 9, independent technical experiments). Source data are provided as a Source Data file

additional 6 min rinse with BGE was performed at 100 psi. The conductive line was rinsed with BGE for 3 min at 29 psi. Before analysis, all samples were diluted with leading electrolyte (AmAC at pH 4.0, in a final concentration of 100 mM). Injection of the samples was performed hydro-dynamically, by applying 5 psi for 60 s, corresponding to 6.8% of the total capillary volume (43 nL), unless stated otherwise. After each sample injection, a BGE post plug was injected by applying 0.5 psi for 25 s (0.3% of the capillary volume). For each analysis a constant flow was established, by applying 0.5 psi and 20 kV over the capillary with a constant temperature of 20 °C.

The CESI 8000 system was coupled to an Impact HD UHR-QqTOF-MS (Bruker Daltonics) via a sheathless CE-ESI-MS interface (Sciex) which allowed optimal alignment between the capillary spray tip and the front of the nanospray shield (Bruker Daltonics). All experiments were performed in positive-ionization mode and a stable electrospray was obtained by generating an electrical field between the CE (ground potential) and a negatively charged spray shield (between −1100 and −1300 V; for a schematic overview see Supplementary Fig. 9). For all analysis, the temperature and flow rate of the drying gas were set at 100 °C and 1.2 L min⁻¹, respectively. To minimize the in-source decay, the collision cell energy as well as the quadrupole ion energy were set at 3.0 eV and the pre-pulse storage was set at 15.0 µs. For the analysis with a dopant enriched nitrogen (DEN) gas, MeCN was used as dopant (ca. 4%, mole percent) at 0.2 bar. In addition, an in-house made polymer cone was attached onto the porous tip housing to enable the usage of the DEN-gas[31]. Fragmentation was performed at 1.00 Hz on the three most abundant precursor ions in a range of *m/z* 150–2000 with a minimum intensity of 4548. Depending on the *m/z* values, the precursor ions were isolated with a width of 8–10 Th. The collision energies were set as a linear curve in a *m/z* dependent manner, ranging from 55 eV at *m/z* 700 to 124 eV at *m/z* 1800 for all charge states (1–5), applying a basic stepping mode with collision energies of 100% (80% of the time) or 50% (20% of the time).

**Sensitivity assessment for the CE-ESI-MS method**. Two *N*-glycan standards (H3N4 and H5N4) were dissolved in water and mixed in a final concentration of 3333 fmol µL⁻¹ for each glycoform. The mixture was diluted 10, 20, 100, 200, 1000, 2000 and 10,000 times and three replicates of 3 µL of each of the dilutions (10,000, 1000, 500, 100, 50, 10, 5 and 1 fmol of each glycan) were subjected to EEA and HILIC purification, and were eluted in 10 µL MQ. The replicates were dried and labelled with GirP following the optimized protocol as described above, and analysed by CE-ESI-MS.

**CE-ESI-MS(/MS) data processing**. Raw CE-ESI-MS data were calibrated prior to data analysis using a minimum of five signals of the identified *N*-glycan compositions (Supplementary Data 1) with Data Analysis 4.2 (Build 395, Bruker Daltonics). The data were manually screened for *N*-glycan compositions based on their exact

mass, migration order and previous described structures in literature, which resulted in 167 *N*-glycan compositions, including differently linked sialic acids (Supplementary Data 1)[5,12,16,24–28]. Fragmentation spectra were acquired for 49% of the identified *N*-glycan variants of TPNG (Supplementary Fig. 6). After converting the raw files into .mzXml files, targeted data analysis was performed using an adapted version of LaCyTools v1.0.1 build 8[32]. Prior to automated peak integration, all electropherograms were aligned based on 11 glycan peaks that were confirmed by tandem MS and that covered the complete migration range (34.5–39.1 min). An alignment time window of 50 s and an *m/z* window of 0.02 Th were used (Supplementary Figs. 10 and 11). Subsequently, for each glycan composition, the area was integrated of isotopic peaks covering at least 95% of the theoretical isotopic pattern and the background was subtracted based on the local background calculations. *N*-glycan compositions were included for further data analysis, when, in at least two repeated experiments per condition, their mass accuracy was between ±10 ppm for the inter- and intraday experiments or ±20 ppm for all other analysis, their isotopic pattern did not deviate more than 20% from the theoretical isotopic pattern and their S/N was above 9. This resulted in 118 quantified *N*-glycan compositions in the TPNG analysis (Supplementary Data 1). The absolute abundance of the lowest abundant quantified glycoforms was estimated based on the 24 most abundant plasma glycoprotein concentrations and their glycosylation sites as reported by Clerc et al.[33]. We assumed a concentration of 0.64 mM released glycans in plasma, and a starting amount of 0.2 µL plasma. In addition, the sensitivity assessment based on the two *N*-glycan standards was evaluated in the condition where both H3N4 and H5N4 still passed the above stated quality criteria.

**MALDI-TOF-MS analysis**. After sialic acid stabilization and cotton HILIC SPE purification, the released *N*-glycans and the sialyllactose standards were prepared for MALDI-TOF-MS analysis by applying 1 µL of the HILIC eluate on an AnchorChip 800/384 TF MALDI target (Bruker Daltonics, Bremen, Germany), together with 1 µL 5 mg mL⁻¹ super-DHB in 50% MeCN with 1 mM NaOH. The samples were left to dry by air. To assess GirP labelling efficiency, 4 µL of HILIC eluates of the GirP-labelled *N*-glycans, were applied on the MALDI target together with 1 µL of the super-DHB matrix. The samples were left to dry by air and re-crystallized with 0.2 µL EtOH.

MALDI-TOF-MS analysis was performed on a Bruker Daltonics UltraFlextreme, equipped with a Smartbeam-II laser and controlled by Flexcontrol 3.4 software. The instrument was calibrated with the peptide calibration standard (Bruker Daltonics) prior to analysis of the samples. MS measurements were performed in reflectron positive ion mode, using an acceleration voltage of 25 kV after 140 ns delayed extraction. Spectra were recorded between *m/z* 1000 and *m/z* 5000 (for the *N*-glycans) or *m/z* 100 and *m/z* 1500 (for the sialyllactose standards). One-hundred shots per raster spot were collected in a random walk, to a total of 10,000 laser shots at a laser frequency of 1000 Hz. The laser power was adjusted to

the maximum at which isotopic resolution was maintained and the isotopic pattern of the highest signal was not disturbed.

**MALDI-TOF-MS data processing**. Raw MALDI-TOF-MS data were converted to .xy files by flexAnalysis Batch Process (Bruker Daltonics). Targeted data analysis was performed using MassyTools v0.1.8.1[34]. Plasma $N$-glycan data were internally recalibrated based on a minimum of six high abundant signals and data integration was performed using a manually determined list of $N$-glycan compositions based on the CE-ESI-MS data as described above, Supporting Information and literature[12]. For the sialyllactose standards, the glycan composition H2S1 with a lactonized, amidated or ethyl-esterified sialic acid was integrated. For each composition, the area was integrated of isotopic peaks covering at least 95% of the theoretical isotopic pattern and the background was subtracted based on the local background calculations. $N$-glycan compositions were included for further data analysis, when, in at least two repeated experiments per condition, their mass accuracy was between ± 20 ppm, their isotopic pattern did not deviate more than 20% from the theoretical isotopic pattern and their signal-to-noise ratio (S/N) was above 9. GirP labelling efficiency was determined based on the seven glycoforms that were reliable quantified (using the same quality criteria for analytes as stated above) in both their labelled (detected as $[M]^+$) and their non-labelled form (detected as $[M + Na]^+$; Supplementary Fig. 3). Part of the GirP-labelled glycoforms lost the pyridinium group (the cationic moiety) of their label during MALDI and were detected as $[M + Na]^+$. However, as this was a MALDI-specific effect, not observed in the CE-MS experiments, the glycoforms with fragmented label were considered part of the labelled fraction. The labelling efficiency was calculated by dividing the labelled fraction (full label + fragmented label) over the sum of the unlabelled and the labelled signals per analyte. Subsequently, per analyte, the average of the labelling efficiency was calculated over the replicates. Finally, average and standard deviation were calculated over the average labelling efficiencies of the seven selected analytes.

**Reporting summary**. Further information on research design is available in the Nature Research Reporting Summary linked to this article.

## Data availability
The raw mass spectrometric data files that support the findings of this study are available in MassIVE in .mzXML and .xy format, with the identifier MSV000083478 [https://doi.org/10.25345/C5061Z][35]. The source data underlying Fig. 4, Supplementary Figs. 1, 2, 5, 7 and 10, and Supplementary Tables 1 and 4 are provided as a Source Data file. A reporting summary for this Article is available as a Supplementary Information file. All other data supporting the findings of this study are available from the corresponding author on reasonable request.

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

## Acknowledgements
We thank Gerda C.M. Vreeker for her valuable input on the annotation of the *N*-glycan structures. We thank Ludger Ltd. for providing the glycan standards. This research was supported by the European Union's Seventh Framework Programme HighGlycan (grant agreement no. 278535), Cure for Cancer Foundation, Astellas Pharma B.V. and SCIEX.

## Author contributions

G.L., N.H., P.M., S.W., M.F. and J.N. performed the experiments. G.L., N.H. and M.W. conceptually designed the work and wrote the manuscript. D.F. assisted in the experimental design. All authors read and commented on the manuscript.

## Additional information

**Competing interests:** The authors declare no competing interests.

