## [Peer Review File · Nature Communications]

Reviewers' comments:

Reviewer #1 (Remarks to the Author):

The significance is high as sialic acids are challenging to differentiate and quantify, but are relevant to different pathological states. As this field advances, each new strategy provides insight into the abundance of N-glycan motifs, but also brings to light different biases in sample preparation and analyses. As a result, each new method or methodological advance (including this report) is an important step forward. These authors have a strong track record in glycan analyses and the current manuscript summarizes new technology that builds upon, but differs, from their prior work. The scientific advances outlined in this report address barriers in determining the ratio of sialic acid linkages. The reaction to differentiate sialic acid linkages, which is a major innovation in the report applied to a relevant biological sample, is thoroughly optimized and reported in great detail.

The authors have provided a highly detailed methods sections in the supporting information. It would be helpful if the authors might also include catalog numbers and lot numbers (when available) for chemicals (especially for biological reagents). The authors have also provided substantial instructions regarding the instrument operation. Could the authors clarify how the voltage between -1100 V and -1300 V is applied to the glass capillary? If the authors feel it is practical and if the vendor does not object to reproducing images of the software and/or instrument components, would it be feasible to include images of the program settings and components?

Reviewer #2 (Remarks to the Author):

Capillary electrophoresis-mass spectrometry (CE-MS) is a technique that seems perpetually on the verge of making an impact on glycomics. With greater peak capacity, CE can separate glycan isomers that are not separable by MS-compatible liquid chromatography methods. The problem has always been with the difficulty of achieving reproducible performance. There are many publications showing impressive results that were obtained with great effort by skilled operators that had little impact because they could not be replicated by other laboratories.

The present work demonstrates a CE-MS method that may be a significant advance. The authors have used a published coating method to achieve capillary conditions that are consistent from run-

to-run. They describe a capillary preparation procedure that appears to be quite effective for glycan separations.

The sensitivity assessment for the CE-ESI-MS method does not help readers understand the minimum quantity of glycans required for successful analysis. The authors should state clearly what glycan volumes and concentrations are required for derivatization and subsequent CE-ESI-MS.

Additional supplementary information should be provided so that readers can evaluate the method. In particular, extracted CE-mass spectra should be shown to illustrate the extent to which cation adducts complicate the mass spectral profiles. Such adducts complicate use of tandem mass spectrometry for glycans.

The stated interday variability of RSD 9.4% depends on the ability to compensate for changes in migration time in the CE electropherograms. It is therefore essential that both raw CE electropherograms and extracted ion electropherograms used for the repeatability studies be shown. Since the reported repeatability depends on the ability to align the electropherograms, the data should be included in the supplemental information.

All mass spectral data should be shared via a public proteomics or glycomics server.

Guinevere S.M. Lageveen-Kammeijer, MSc
Center for Proteomics and Metabolomics,
Leiden University Medical Center (LUMC)
P.O. Box 9600, 2300 RC Leiden, The Netherlands
E-mail: g.s.m.kammeijer@lumc.nl
Tel: +31-71-52-69384

Dear Reviewer,

We would like to thank you for considering our manuscript entitled: **“Highly sensitive CE-ESI-MS analysis of N-glycans from complex biological samples”** (by Guinevere S.M. Lageveen-Kammeijer, Noortje de Haan, Pablo Mohaupt, Sander Wagt, Mike Filius, Jan Nouta, David Falck and Manfred Wuhrer) for publication in Nature Communications.

We tried to fully address the comments and feel that our manuscript substantially improved. Please find below a point-to-point response to the concerns that were raised. In the revised manuscript we highlighted changes with track changes, colored red for deletions and colored green for insertions.

The submitted files include the Supplementary information for Review Only (.docx) (with highlighted changes), the Main Manuscript (.docx), 4 Main Figures (.pdf), Source Data (.xlsx), Supplementary Information with 11 Supporting Figures and 3 Supporting Tables (T-1, T-2 & T-4) (.pdf), 1 Supplementary Table (T-3) (.xlsx).

We remain at your disposal in case of any further inquiries you may have and we look forward to hearing from you.

Yours truly,

Prof. Dr. Manfred Wuhrer, Dr. Noortje de Haan and Guinevere S.M. Lageveen-Kammeijer, MSc

RESPONSE TO THE REVIEWERS' COMMENTS

REVIEWER #1

We thank the reviewer for his/her valuable comments regarding this study. We have addressed them below.

COMMENTS:

The significance is high as sialic acids are challenging to differentiate and quantify, but are relevant to different pathological states. As this field advances, each new strategy provides insight into the abundance of N-glycan motifs, but also brings to light different biases in sample preparation and analyses. As a result, each new method or methodological advance (including this report) is an important step forward. These authors have a strong track record in glycan analyses and the current manuscript summarizes new technology that builds upon, but differs, from their prior work. The scientific advances outlined in this report address barriers in determining the ratio of sialic acid linkages. The reaction to differentiate sialic acid linkages, which is a major innovation in the report applied to a relevant biological sample, is thoroughly optimized and reported in great detail.

1. The authors have provided a highly detailed methods sections in the supporting information. **It would be helpful if the authors might also include catalogue numbers and lot numbers (when available) for chemicals (especially for biological reagents).**

We now included all catalogue numbers of all chemicals used in this study in the methods section (lines 132-156). In addition, the lot numbers of the plasma pool and all glycan standards were added (lines 152-154).

2. The authors have also provided substantial instructions regarding the instrument operation. **Could the authors clarify how the voltage between -1100 V and -1300 V is applied to the glass capillary? If the authors feel it is practical and if the vendor does not object to reproducing images of the software and/or instrument components, would it be feasible to include images of the program settings and components?**

To achieve ESI, an electrical field is created between the CE (ground potential) and the negatively charged spray shield of the MS (between -1.1 and -1.3 kV). A schematic overview of the instrument components of our setup is provided in the new Supplementary Figure 11. This Figure is referred to in the Methods section (line 232).

REVIEWER #2

We thank the reviewer for the time invested in carefully checking our manuscript. We have addressed the points raised by the reviewer as detailed below.

COMMENTS:

Capillary electrophoresis-mass spectrometry (CE-MS) is a technique that seems perpetually on the verge of making an impact on glycomics. With greater peak capacity, CE can separate glycan isomers that are not separable by MS-compatible liquid chromatography methods. The problem has always been with the difficulty of achieving reproducible performance. There are many publications showing impressive results that were obtained with great effort by skilled operators that had little impact because they could not be replicated by other laboratories.

The present work demonstrates a CE-MS method that may be a significant advance. The authors have used a published coating method to achieve capillary conditions that are consistent from run-to-run. They describe a capillary preparation procedure that appears to be quite effective for glycan separations.

1. **The sensitivity assessment for the CE-ESI-MS method does not help readers understand the minimum quantity of glycans required for successful analysis. The authors should state clearly what glycan volumes and concentrations are required for derivatization and subsequent CE-ESI-MS.**

We reckon that glycan concentration and sample volumes used are valuable information for the reader when evaluating the results of the sensitivity assessment. The 5 fmol starting amount was present in a 1.67 fmol/ μ L solution of which 3 μ L was used for the first step of the reaction (the derivatization of the sialic acids). Details on this can be found in the methods section (Sensitivity assessment for the CE-ESI-MS method, lines 242-249) and Supplementary Table 2, but are now also added to the Results (lines 87-90).

Additionally we added the information on volumes used for the analysis of the plasma glycans to the Results (lines 93-95). Briefly, we started the derivatization with 1 μ L glycan release, containing the equivalent of 0.2 μ L PNGase F treated plasma. Eventually, the equivalent of 0.1 nL PNGase F treated plasma was injected into the CE-ESI-MS setup.

2. Additional supplementary information should be provided so that readers can evaluate the method. **In particular, extracted CE-mass spectra should be shown to illustrate the extent to which cation adducts complicate the mass spectral profiles. Such adducts complicate use of tandem mass spectrometry for glycans.**

While glycans are indeed known to be prone to form various adducts with cations in MS, this was hardly observed in our measurements (< 1%). We almost exclusively observed singly

charged species $[M]^+$ (charged introduced by reducing end label) and protonated higher charged species (e.g. $[M+H]^{2+}$). This was exemplified by displaying the signals of the highest abundant glycoform: H5N4S_{2,6}2, and a high abundant tri-antennary glycan: H6N5S_{2,3}1S_{2,6}2 in the new **Supplementary Figure 8**. We referred to this Figure in the Results of the Main Manuscript (lines 112-114).

3. The stated interday variability of RSD 9.4% depends on the ability to compensate for changes in migration time in the CE electropherograms. **It is therefore essential that both raw CE electropherograms and extracted ion electropherograms used for the repeatability studies be shown. Since the reported repeatability depends on the ability to align the electropherograms, the data should be included in the supplemental information.**

We would like to thank the reviewer for pointing out that the information on the alignment of the data was missing from the Methods section, we added the details in “CE-ESI-MS(/MS) data processing” (lines 258-261).

Moreover, it is indeed true that the alignment is essential for automated data integration. In the new **Supplementary Figure 10** we now demonstrate that the repeatability in terms of quantitation is not dependent on the alignment of the electropherograms (referred to in the legend of Figure 4 – line 445). Namely, the integration of manually assigned peaks without alignment versus automatic integration results in very similar values for the repeatability assessment (i.e. a median RSD of 9.5% for the automated version and of 9.3% after manual peak picking without alignment).

Additionally, **Supplementary Figure 9** has been added to visualize the effect of the alignment on the migration times (raw data versus aligned data) using MZmine v2.30 (line 261 and in the legend of Figure 4 – line 445). Electropherograms were aligned in a straightforward manner using the open source software LacyTools (Jansen et al. 2016).

4. **All mass spectral data should be shared via a public proteomics or glycomics server.**

The raw mass spectrometric data files that support the findings of this study are now available in MassIVE in .mzXML (CE-ESI-MS) and .xy (MALDI-TOF-MS) format, with the identifiers MSV000083478 and DOI:10.25345/C5061Z.

Moreover, a data availability statement has been added to the manuscript (lines 316-319): “The raw mass spectrometric data files that support the findings of this study are available in MassIVE in .mzXML and .xy format, with the identifiers MSV000083478 and DOI:10.25345/C5061Z.” The source data underlying Fig. 4, Supplementary Figs. 1, 2, 5, 7 and 10, and Supplementary Table 1 are provided as a Source Data file.”

REVIEWERS' COMMENTS:

Reviewer #2 (Remarks to the Author):

The authors have addressed all comments raised in the first review.